# Validity of self-reported hysterectomy: a prospective cohort study within the UK Collaborative Trial of Ovarian Cancer Screening (UKCTOCS)

Aleksandra Gentry-Maharaj,[1] Henry Taylor,[1] Jatinderpal Kalsi,[1] Andy Ryan,[1] Matthew Burnell,[1] Aarti Sharma,[1] Sophia Apostolidou,[1] Stuart Campbell,[2] Ian Jacobs,[1,3] Usha Menon[1]

[1]Gynaecological Cancer Research Centre, Women's Cancer, Institute for Women's Health, University College London, London, UK
[2]Create Health Clinic, London, UK
[3]Faculty of Medical and Human Sciences, University of Manchester, Manchester

Correspondence to
Dr Aleksandra Gentry-Maharaj;
a.gentry-maharaj@ucl.ac.uk

## ABSTRACT

**Objective:** To evaluate the validity of self-reported hysterectomy against the gold standard of uterine visualisation using pelvic ultrasound.

**Design:** Prospective cohort study.

**Setting:** UK Collaborative Trial of Ovarian Cancer Screening (UKCTOCS) based in 13 National Health Service (NHS) Trusts in England, Wales and Northern Ireland.

**Participants:** Between April 2001 and October 2005, 48 215 postmenopausal women aged 50–74 randomised to the ultrasound screening arm of UKCTOCS underwent the first (initial) scan on the trial.

**Interventions:** At recruitment, the women completed a recruitment questionnaire (RQ) which included previous hysterectomy. The sonographer asked each woman regarding previous hysterectomy (interview format, IF) prior to the scan. At the scan, in addition to ovarian morphology, endometrial thickness (ET)/endometrial abnormality were captured if the uterus was visualised at the scan.

**Outcome measures:** Self-reported hysterectomy at RQ or IF was compared to ultrasound data on ET/endometrial abnormality (as surrogate uterine visualisation markers) on the first (initial) scan.

**Results:** Of 48 215 women, 3 had congenital uterine agenesis and 218 inconclusive results. The uterus was visualised in 39 121 women. 8871 self-reported hysterectomy at RQ, 8641 at IF and 8487 at both. The uterus was visualised in 39 123, 39 353 and 38 969 women not self-reporting hysterectomy at RQ, IF or both. Validity, sensitivity, specificity, positive predictive value and negative predictive value of using RQ alone, IF or both RQ/IF were 99.6%, 98.9%, 99.7%, 98.9% and 99.7%; 98.9%, 98.4%, 99.1%, 95.9% and 99.7%; 99.8%, 99.6%, 99.9%, 99.4% and 99.9%, respectively.

**Conclusions:** Self-reported hysterectomy is a highly accurate and valid source for studying long-term associations of hysterectomy with disease onset.

**Trial registration** International Standard Randomised Controlled Trial Number (ISRCTN)—22488978

## Strengths and limitations of this study

- In our study of 48 000 women, we show a very high validity of reporting of hysterectomy status by questionnaire or interview compared to ultrasound scan data.
- This is the first report comparing self-reported hysterectomy status to the gold standard of ultrasound scanning.
- It informs epidemiologists/clinical researchers that self-reported hysterectomy is a reliable data source when assessing its associations with disease risk.
- The scan form did not specifically capture whether the uterus was seen or not and the reason for it.

## INTRODUCTION

Hysterectomy is one of the most commonly performed gynaecological operations.[1] Hysterectomy rates among postmenopausal women are 19%[2] to 25%[3] in the UK and up to 48% in the USA.[4] Most women undergo hysterectomy for benign gynaecological conditions (uterine fibroids, excessive bleeding, endometriosis and uterine prolapse).[4,5] Over the past two decades, an increasing body of literature has identified the impact of hysterectomy on disease risk. It has consistently been associated with reduced risk for ovarian cancer with a recent meta-analysis reporting a relative risk of ovarian cancer of 0.74 among women who had undergone hysterectomy.[6] Further associations with increased risk of cardiovascular disease[7] and reduced risk of breast cancer[8] have also been noted. In such studies, hysterectomy status is almost invariably ascertained by asking women whether they have had a hysterectomy or removal of the womb. However, there is a paucity of studies which analyse the accuracy

of self-reporting of hysterectomy as a reliable method of data collection. We are aware of five small studies involving 79–455 women who have verified the validity of self-reported hysterectomy by review of medical notes or electronic records (table 1).[9–13] Four of the five studies report an agreement of 91% or over with one study of 452 women reporting a validity of only 66%, as the hospital records could not confirm that the procedure had been undertaken.

In the ultrasound screening group of the UK Collaborative Trial of Ovarian Cancer Screening (UKCTOCS),[2] 50 639 women were randomised to an ultrasound scan of the pelvis following completion of a baseline questionnaire which included a question on hysterectomy. In this study, we report on the agreement between self-reported hysterectomy and ultrasound confirmation of the presence/absence of a uterus.

## METHODS
### Study design and subjects
UKCTOCS is a randomised controlled trial designed to assess whether screening for ovarian cancer has an impact on mortality from the disease. Between April 2001 and October 2005, 202 638 apparently healthy women aged 50–74 were recruited to the trial through 13 regional trial centres located in National Health Service (NHS) Trusts in England, Wales and Northern Ireland. Women completed a baseline questionnaire on lifestyle and medical history, which included a question on hysterectomy: 'Have you ever had a hysterectomy (removal of the womb)?' (questionnaire format, recruitment questionnaire (RQ)). The women were then randomised to screening either with (1) transvaginal ultrasound (ultrasound group, USS; n=50 639); (2) CA125 interpreted by the Risk of Ovarian Cancer (ROC) algorithm (multimodal group, MMS; n=50 640) or (3) control (no screening, n=101 359).[14]

The women randomised to the ultrasound arm of UKCTOCS were invited to attend for a transvaginal scan (TVS). At this visit, they were again questioned by the ultrasonographer about previous hysterectomy prior to the scan (interview format, IF). At each scan, in addition to data regarding the adnexae, the sonographer recorded the endometrial thickness (ET) and presence of any endometrial or uterine abnormalities. They also recorded any other relevant data as 'text' in the notes section.

### Interpretation of the scan findings
The scan data were used to ascertain the presence or absence of a uterus. The uterus was deemed to be present if ET was measured or uterine abnormality, for example, 'fibroids' was noted. In addition, the uterus was considered to be present if the sonographer noted an abnormal uterine position, the presence of an intra-uterine device or made specific comments regarding uterine visualisation in the notes section. The uterus was

**Table 1** Studies comparing validity of self-reported hysterectomy compared to hospital/medical records

| Author (year) | Study design | Self-reported hysterectomy compared to source | N, reporting hysterectomy | N, who had hysterectomy status confirmed | N, where confirmation was impossible | Quoted validity of self report (%) |
|---|---|---|---|---|---|---|
| Brett and Madans (1994)[12] | Cross-sectional | Hospital records | 452 | 298 | 33 | 66 |
| Green et al (1997)[10] | Case–control | Medical records, surgeon's report, or GP | 206 | 127 | 73 | 96 |
| Phipps and Buist (2009)[9] | Cross-sectional | Surgical pathology report (10%), imaging reports (35%) and transcribed medical history (48%) | 79 | 72 | 5 | 91 |
| Colditz et al (1987)[11] | Cross-sectional | Medical records | 212 | 199 | 12 | 99 |
| (Horwitz (1986)[13] | Case–control | Medical records | 455 | 440 | 34 | 97 |

GP, general practitioner.

deemed to be absent if there were no ET measurements or if none of the above were captured as text in the notes field. Scans with poor visualisation of the pelvis or where the procedure was abandoned due to discomfort were classified as 'inconclusive' and these women were excluded from the analysis.

## Data analysis

Self-reported hysterectomy status on the questionnaire or at the interview was compared to ultrasound findings of the presence or absence of a uterus.

We estimated sensitivity as the proportion of women with an absent uterus confirmed on scan, who self-reported hysterectomy on RQ, at IF or both. Specificity was calculated as the proportion of women with a uterus confirmed on scan, who self-reported not having a hysterectomy. Positive predictive value (PPV) was calculated as the proportion of women who did self-reported hysterectomy, where the scan confirmed the absence of a uterus. Negative predictive value (NPV) was calculated as the proportion of women who did not self-report hysterectomy, where the scan confirmed the presence of a uterus. Validity was calculated as the proportion of women who were accurate in their reporting (self-reported hysterectomy with uterus absent and those who did not report it with uterus present) out of all women.

We further evaluated whether age at randomisation or socioeconomic status as derived from the Index of Multiple Deprivation (IMD) could affect the validity of reporting.

## RESULTS

Between 2001 and 2005, 50 639 women randomised to the ultrasound arm of UKCTOCS completed the RQ which included self-reported hysterectomy. Of them, 48 215 underwent the first (initial) scan on the trial. Of these women, 42 401 had a TVS, 4325 a transabdominal

(TA) scan and 1489 had both performed at the same time. The median time from randomisation to scan was 2.7 weeks (IQR, 1.7–4.0). The uterus was reported to be present on the scan report in 38 221 and absent in 9398 women. The latter included three women who had congenital uterine agenesis. In 593 women, the ultrasound report was inconclusive with regard to the presence or absence of a uterus on this initial scan. Of these women, 242 had a TVS, 299 had a TA and 52 had both scans performed. As the 593 women went on to have 2–11 annual scans as part of the trial, these and all the other scans they have had were then queried to ascertain the presence or absence of a uterus. On subsequent scans, in 353 the presence of a uterus was recorded, 22 had the absence of a uterus clearly stated and 218 remained inconclusive due to difficulty in visualising the pelvis (figure 1). Excluding the latter and women with congenital uterine agenesis, 47 994 women were therefore eligible for the analysis with ultrasound confirming the presence of a uterus in 39 121 and the absence of a uterus in 8873.

Of the 47 994 women eligible for analysis, 8871 self-reported hysterectomy at recruitment (RQ) and this was later confirmed on scan for 8771 women. Of the 39 123 women who did not report hysterectomy on RQ, 39 021 had a uterus visualised at scan. In 102 women who did not report hysterectomy, a uterus was not found despite good visualisation of the pelvis.

At the interview (IF) prior to the scan, of the 47 994 women, the ultrasonographer recorded that 8641 reported hysterectomy and 8505 had it confirmed by scan data. Of the 39 353 women who did not report hysterectomy on IF, 38 985 had a uterus visualised at scan. Of the 8871 women who self-reported hysterectomy on the RQ, 8487 also reported it when interviewed immediately prior to the scan (IF) and 8457 had it confirmed by scan data. From the 39 123 women who did not report hysterectomy on RQ, 38 969 did not report it at

**Figure 1** UKCTOCS participants randomised to the ultrasound group who underwent the initial scan on the trial which confirmed the presence or absence of a uterus. IF, interview format; RQ, recruitment questionnaire; UKCTOCS, UK Collaborative Trial of Ovarian Cancer Screening.

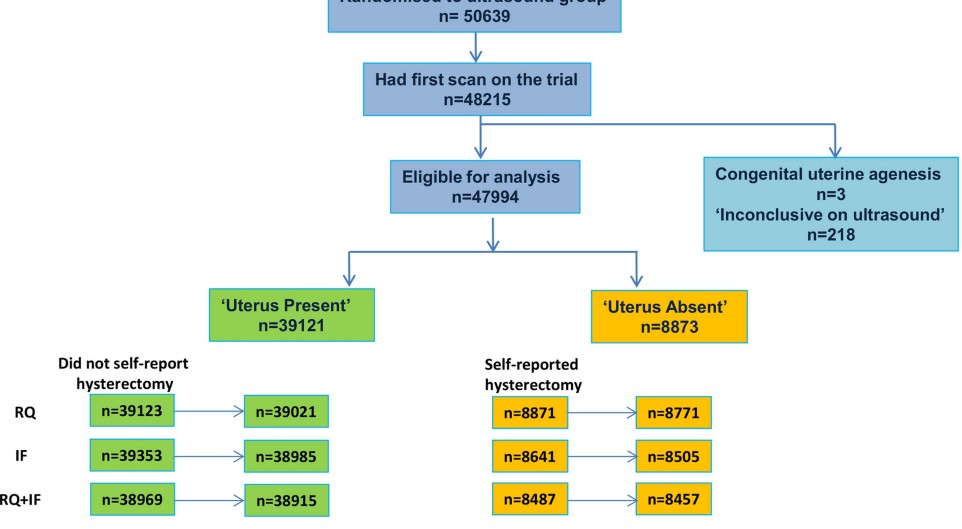

IF either and the uterus was seen in 38 915. Seventy of the 384 women who self-reported hysterectomy on RQ but not at IF had a uterus visualised, while in 106 of the 154 women who did not report hysterectomy on RQ but did at IF the uterus was present. Fifty-four women had uterine absence noted at the scan despite not reporting having had a hysterectomy at either the RQ or IF.

### Performance characteristics

The validity, sensitivity, specificity, PPV and NPV of using RQ alone were 99.6%, 98.9%, 99.7%, 98.9% and 99.7%. Using the interview data alone, these were 98.9%, 98.4%, 99.1%, 95.9% and 99.7%. A McNemar test for paired proportions showed that RQ had greater validity than IF (99.6% vs 98.9%; p<0.0001). Consistent self-reporting at both RQ and IF resulted in validity, sensitivity, specificity, PPV and NPV of 99.8%, 99.6%, 99.9%, 99.4% and 99.9%, respectively (table 2).

### Correlation with age and IMD score

The women who accurately reported the presence or absence of a uterus (whether on RQ, IF or at both RQ and IF) were younger (median of 60.5, IQR 56.0–66.0) compared to those who were inaccurate in their reporting (median of 63.5, IQR 56.6–69.2) for RQ, 61.9 (IQR 56.5–67.7) for IF, 64.6 (IQR 57.9–69.1) for RQ and IF (figure 2A). A Mann-Whitney test of the difference in rankings was significant at the 1% level (p=0.0006). The IMD score in those accurately reporting the presence or absence of a uterus was 15.0 (IQR 8.9–26.3) compared to 17.6 (IQR 10.9–29.4) for RQ, 16.4 (IQR 9.6–26.9) for IF and 18.2 (IQR 9.0–28.3) for both RQ and IF in those who were inaccurate (figure 2B). A Mann-Whitney test showed some evidence of a difference in IMD score rankings (p=0.0214).

### DISCUSSION

To our knowledge, this is the largest study until now to assess the validity of self-reported hysterectomy by comparing it to a gold standard of a pelvic ultrasound scan performed at a median of 2.7 weeks following questionnaire reporting. Self-reporting of hysterectomy was correct in 99.6% of cases, indicating that self-reporting

(as captured in epidemiological studies) is an accurate measure of hysterectomy status. The validity of capturing hysterectomy using the questionnaire or interview was similar (99.6% vs 98.9%) though statistically significant. As the data on hysterectomy collected in most studies are usually ascertained through self-reporting, we feel that the data presented here can inform epidemiologists about the reliability of self-reporting of this variable.

The strengths of our study are (1) the size, (2) prospective design and (3) the fact that UKCTOCS participants were recruited from 13 centres across England, Wales and Northern Ireland through invitation from population registries—they are therefore as representative of the general female population aged 50–74 as is possible in clinical research.[14] We have previously observed a 'healthy volunteer effect' in the trial,[15] so the women participating in UKCTOCS may be more health-conscious and aware of medical procedures than the general population, therefore being highly accurate in reporting. All women randomised to the ultrasound arm of UKCTOCS who attended for the initial scan were included in the analysis. The scans were in the majority transvaginal, performed within 3 weeks of self-reporting, with all ultrasonographers using similar ultrasound machines and following a uniform protocol and data capture form to record findings.[2] Where there was a discrepancy or where the ultrasound records were inconclusive, all serial scans undertaken in the women during screening were analysed. Furthermore, two types of self-reporting (questionnaire vs interview) were assessed and their validity compared.

In our study, nearly 48 000 women were included in the analysis with prospective evaluation of hysterectomy. Previous studies have been retrospective with a medical note review used to confirm hysterectomy status. Logistical issues have meant that these studies were relatively small. They frequently encountered difficulties in acquiring notes because they relied on patients remembering details of hospitals where a procedure, which potentially occurred decades ago, was undertaken and there was often poor recording of events.[9 10] In the multicentre Study of Women Across the Nation (SWAN), of the 3302 participants, 239 reported hysterectomies of which for only 165 (69.3%) of the women the procedure could be confirmed. The remaining 74 could not be reviewed due to consent issues, loss to follow-up or unavailability of medical records.[16] In addition, some studies have used self-reported hysterectomy transcribed in medical histories as part of validation.[9] This has resulted in agreement of self-reported hysterectomy with medical records ranging from 66% to 99%.[9–13] A methodology that avoids several of these difficulties is to identify a cohort of women who have undergone hysterectomy using electronic outpatient records and subsequently investigate their self-reported hysterectomy status. Phipps and Buist,[9] using this approach, found that of 1935 women with confirmed hysterectomy, 1757 self-reported correctly, giving a sensitivity of 91%.

**Table 2** Validity, sensitivity, specificity, PPV and NPV of self-reporting hysterectomy compared to the gold standard of ultrasound scanning

|  | Self-reported hysterectomy | | |
|---|---|---|---|
|  | RQ (%) | IF (%) | RQ and IF (%) |
| Validity | 99.6 | 98.9 | 99.8 |
| Sensitivity | 98.9 | 98.4 | 99.6 |
| Specificity | 99.7 | 99.1 | 99.9 |
| PPV | 98.9 | 95.9 | 99.4 |
| NPV | 99.7 | 99.7 | 99.9 |

IF, interview format; NPV, negative predictive value; PPV, positive predictive value; RQ, recruitment questionnaire.

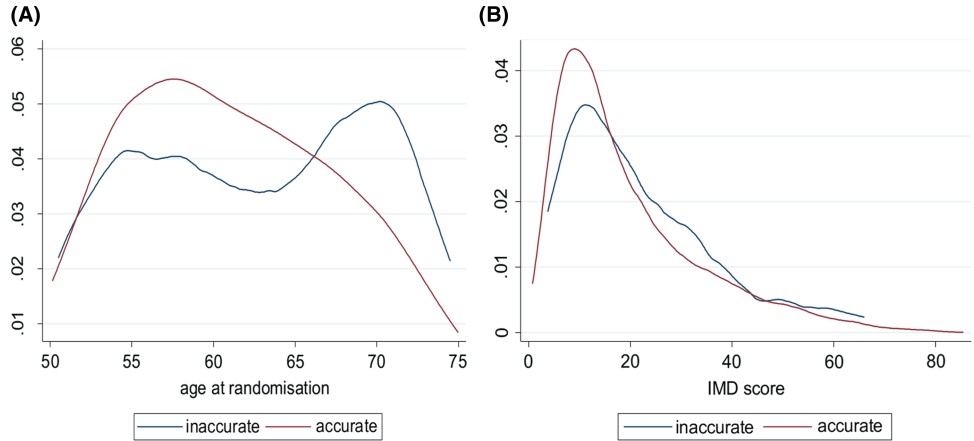

**Figure 2** (A) Kernel density plots of age at randomisation and (B) Index of Multiple Deprivation (IMD) for those who were accurate in their reporting versus those who were inaccurate.

Our study showed validity of self-reported hysterectomy of 99% in the UK women from the general population, aged 50–74. These figures are similar to the 100% accuracy rates reported in a subgroup of 200 women, for whom medical records could be obtained, participating in the Nurses' Health Study.[11] However, this cohort was composed entirely of women with a nursing background who are well aware of the procedure, making it difficult to extrapolate to the wider population. Our results indicate that self-reporting of hysterectomy is highly accurate in women from the UK, and we feel that our findings are applicable to women from other developed countries. However, it is possible that the validity of self-reporting of hysterectomy may be lower in those societies where there is a stigma attached to having undergone this procedure.

A limitation of our study is that the sonographer was not asked to specifically capture the presence or absence of a uterus. ET or endometrial abnormality were the only uterine details that were recorded.

A small proportion of women reported hysterectomy at recruitment but failed to do so at the scan. The reason for this discrepancy may be that either the sonographer did not question the woman or the woman did not hear the question at the time of the scan.

The women who were accurate in their reporting were younger than those who were inaccurate in their reporting of the presence or absence of a uterus. One of the reasons for this may be due to the procedure being carried out many years, even decades previously.

In our study, 1.2% of women (576 of 47 994) were inconsistent in their self-reporting of hysterectomy status between the RQ and the interview carried out at a median 2.7 weeks later. Similarly, studies examining the agreement of self-reports report a low rate of inconsistency, but these were made several years apart.[17 18] One of the reasons for this may be that the women were more accurate in reporting on the questionnaire as they had more time to fill this in compared to those attending their first scan in a busy clinic. Our data and previous data indicate that capturing data via an interviewer-administered questionnaire is a valid method to capture self-reported medical history.

## CONCLUSION
Our data indicate that self-reported hysterectomy, whether through a questionnaire or interview, is highly accurate and is a reliable data source for studying long-term associations of hysterectomy with disease onset.

**Acknowledgements** The authors are particularly grateful to the women throughout the UK who are participating in the trial and to the entire medical, nursing and administrative staff who work on the UK Collaborative Trial of Ovarian Cancer Screening (UKCTOCS).

**Contributors** UM and AG-M were involved in study design and concept and drafted the manuscript. UM, AG-M and HT did the literature search for this manuscript and prepared the tables. UM, AG-M and MB did the statistical analysis. JK, AR, AS, SA, SC and IJ were involved in the interpretation of data. All authors critically revised the manuscript and approved the final version. UM is the guarantor.

**Funding** UKCTOCS was core funded by the Medical Research Council (grant numbers G9901012, G0801228), Cancer Research UK, and the Department of Health with additional support from the Eve Appeal, Special Trustees of Bart's and the London, and Special Trustees of University College London Hospitals (UCLH) and supported by researchers at the National Institute for Health Research University College London Hospitals Biomedical Research Centre.

**Competing interests** IJ has a consultancy arrangement with Becton Dickinson in the field of tumour markers and ovarian cancer. UM has a financial interest through University College London (UCL) Business and Abcodia Ltd in the third party exploitation of clinical trials biobanks which have been developed through the research at UCL.

**Patient consent** Obtained.

**Ethics approval** The study was approved by the UK North West Multicentre Research Ethics Committees (North West MREC 00/8/34) with site-specific approval from the local regional ethics committees and the Caldicott guardians (data controllers) of the primary care trusts.

**Provenance and peer review** Not commissioned; externally peer reviewed.

**Data sharing statement** No additional data are available.

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
