## [Reviewer comments · BMJ Open]

Some articles will have been accepted based in part or entirely on reviews undertaken for other BMJ Group journals. These will be reproduced where possible.

ARTICLE DETAILS

TITLE (PROVISIONAL)	VALIDITY OF SELF-REPORTED HYSTERECTOMY - PROSPECTIVE COHORT STUDY WITHIN THE UNITED KINGDOM COLLABORATIVE TRIAL OF OVARIAN CANCER SCREENING (UKCTOCS)
AUTHORS	Gentry-Maharaj, Aleksandra; Taylor, Henry; Kalsi, Jatinderpal; Ryan, Andy; Burnell, Matthew; Sharma, Aarti; Apostolidou, Sophia; Campbell, Stuart; Jacobs, Ian; Menon, Usha

VERSION 1 - REVIEW

REVIEWER	Sean Kehoe Univeristy of Birmingham UK
REVIEW RETURNED	02-Dec-2013

GENERAL COMMENTS	This is the largest such study ever reported, and the novel aspect relates not just wiht regard tothe numbers of women involved but also the fact that ultrasound scan has been used as the gold standard. Granted there were some asumptions made - in that if certain comments were not made on the scan report - then the assumption was that the uterus was not in situ. However, this point is clearly made and discussed within the text.
---

REVIEWER	Ray Merrill Brigham Young University, USA
REVIEW RETURNED	26-Dec-2013

GENERAL COMMENTS	This is a nice study verifying the validity of self reported hysterectomy in England and Ireland. I only have a couple comments. First, the results may be limited in generalizability, which should be mentioned in the Discussion. Perhaps some societies may have a stigma attached to hysterectomy, thereby causing less validity in self-reports. Second, likelihood ratio tests have some advantages over the measures of validity used in this study. I am curious why they were not considered?
---

VERSION 1 – AUTHOR RESPONSE

Reviewer #1:

This is the largest such study ever reported, and the novel aspect relates not just with regard to the numbers of women involved but also the fact that ultrasound scan has been used as the gold standard. Granted there were some assumptions made - in that if certain comments were not made on the scan report - then the assumption was that the uterus was not in situ. However, this point is clearly made and discussed within the text.

We would like to thank the reviewer for the comments.

Reviewer #2:

This is a nice study verifying the validity of self-reported hysterectomy in England and Ireland. I only have a couple comments. First, the results may be limited in generalizability, which should be mentioned in the Discussion. Perhaps some societies may have a stigma attached to hysterectomy, thereby causing less validity in self-reports. Second, likelihood ratio tests have some advantages over the measures of validity used in this study. I am curious why they were not considered?

We thank the reviewer for the comments and have now expanded on the generalizability of our findings to other societies. The following sentence has been inserted on page 12 of the discussion: "Our results indicate that self-reporting of hysterectomy is highly accurate in women from the UK and we feel that our findings are applicable to women from other developed countries. However, it is possible that the validity of self-reporting of hysterectomy may be lower in those societies where there is a stigma attached to having undergone this procedure."

Reviewer 2 enquires about likelihood ratio tests which are a means of testing between 2 nested statistical models, fitted my maximum likelihood. Assuming the reviewer refers to the likelihood ratio for diagnostic testing, we feel such a statistic has limited use in this scenario. The LR is defined as the sensitivity divided by 1-specificity, and can be used to assess the added value of a diagnostic test. Specifically, it indicates by what factor a positive test changes the pre-odds of having the condition (say, disease) to the post-odds (i.e after having the positive test).

However, such a test is perhaps not of great use in our study, given that the women reported a hysterectomy, we wonder how would the prior-odds of having had a hysterectomy (which will based on population or sample prevalence) have changed. From a practical point it would also be not especially helpful as it will involve a very large probability divided by a very small probability, given a somewhat uninterpretable large number: for RQ, the $LR = 0.989/0.003 = 330$.

Thank you for considering the revised version of this manuscript.

Conflict of interest: IJ has a consultancy arrangement with Becton Dickinson in the field of tumour markers and ovarian cancer, UM has a financial interest through UCL Business and Abcodia Ltd in the third party exploitation of clinical trials biobanks which have been developed through the research at UCL. None of the other authors have any conflict of interest; no other relationships or activities that could appear to have influenced the submitted work.